# Antibiotic Resistance, Susceptibility Testing and Stewardship in *Helicobacter pylori* Infection

**DOI:** 10.3390/ijms241411708

**Published:** 2023-07-20

**Authors:** Ho-Yu Ng, Wai K. Leung, Ka-Shing Cheung

**Affiliations:** 1School of Clinical Medicine, The University of Hong Kong, Hong Kong, China; nghoyu@connect.hku.hk; 2Department of Medicine, School of Clinical Medicine, The University of Hong Kong, Queen Mary Hospital, 102 Pokfulam Road, Hong Kong, China; 3Department of Medicine, The University of Hong Kong-Shenzhen Hospital, Shenzhen 518053, China

**Keywords:** *H. pylori*, vonoprazan, bismuth, stool based-PCR, next generation sequencing, NGS

## Abstract

Despite the declining trend of *Helicobacter pylori* (*H. pylori*) prevalence around the globe, ongoing efforts are still needed to optimize current and future regimens in view of the increasing antibiotic resistance. The resistance of *H. pylori* to different antibiotics is caused by different molecular mechanisms, and advancements in sequencing technology have come a far way in broadening our understanding and in facilitating the testing of antibiotic susceptibility to *H. pylori*. In this literature review, we give an overview of the molecular mechanisms behind resistance, as well as discuss and compare different antibiotic susceptibility tests based on the latest research. We also discuss the principles of antibiotic stewardship and compare the performance of empirical therapies based on up-to-date resistance patterns and susceptibility-guided therapies in providing effective *H. pylori* treatment. Studies and clinical guidelines should ensure that the treatment being tested or recommended can reliably achieve a pre-agreed acceptable level of eradication rate and take into account the variations in antibiotic resistance across populations. Local, regional and international organizations must work together to establish routine antibiotic susceptibility surveillance programs and enforce antibiotic stewardship in the treatment of *H. pylori*, so that it can be managed in a sustainable and efficient manner.

## 1. Introduction

*Helicobacter pylori* (*H. pylori*) infects approximately 40% of individuals worldwide [1] and is involved in various gastrointestinal diseases such as chronic gastritis, peptic ulcer disease, upper gastrointestinal bleeding [2], gastric cancer (including gastric adenocarcinoma and gastric MALT lymphoma) [3,4,5] and extraintestinal manifestations [6]. All patients infected with *H. pylori* should be treated regardless of clinical manifestations [4]. Eradication of *H. pylori* can restore the normal gastric mucosa [7] and was shown to effectively reduce the development and recurrence of peptic ulcers [8,9], as well as the incidence of gastric cancer [10,11].

Various treatment regimens for *H. pylori* have been developed and their use vary across different geographical regions largely based on local antibiotic resistance patterns. Currently, an acceptable *H. pylori* treatment regimen is defined as one that achieves at least a 90% cure rate [12], though it has been suggested that an optimized regimen should reliably achieve ≥95% cure rates [13]. Treatment regimens currently recommended by various international guidelines are of empirical nature [4,14,15,16]. In light of a substantial decline in efficacy to levels below 80–85% [12,17], triple therapy consisting of a proton pump inhibitor (PPI), clarithromycin, and amoxicillin or metronidazole is usually not recommended as an empirical first-line therapy for *H. pylori* infection, except in areas with a known clarithromycin resistance rate of <15% [4,14,15]. Instead, bismuth-based quadruple therapy, typically consisting of bismuth, PPI, metronidazole and tetracycline, is increasingly recommended as its replacement for first-line treatment [4,14,15] and has been shown to achieve over 90% success rate [18,19]. Bismuth-based quadruple therapy can also be used as a second-line treatment. Another option for second-line treatment is a triple therapy consisting of a PPI, amoxicillin and levofloxacin, but this too faces the problem of decreasing efficacy because of rising resistance to levofloxacin [20].

Antimicrobial stewardship, which is the responsible use of antimicrobials to balance individual needs and long-term societal needs, is increasingly promoted to limit the increasing antibiotic resistance around the world [21]. One important component of antimicrobial stewardship is widely available antibiotic susceptibility testing to provide information for designing optimal treatment regimens. Routine susceptibility testing has been recommended even before first-line treatment in compliance to antibiotic stewardship [4,22]. In particular, the use of molecular-based testing methods, including polymerase chain reaction (PCR)-based assays and next-generation sequencing (NGS), have gained prominence in recent years as an alternative to traditional culture-based testing, each with their own pros and cons. These molecular-based methods can be potentially applied to stool samples obtained through non-invasive means; however, their accuracies, so far, were inconsistent among different studies and need to be studied in more detail [23,24,25,26]. Moreover, challenges such as practicality and cost-effectiveness issues in their use in routine clinical setting should be considered and addressed. In this regard, some studies showed that empirical therapies based on the latest antibiotic resistance patterns could achieve similar efficacy [27,28,29] and be more cost effective [30,31] than susceptibility-guided therapy. More studies are needed to compare the performance of new empirical treatments, such as vonoprazan-based therapies, with susceptibility-guided therapy.

In this review, we will give an overview of the current *H. pylori* antibiotic resistance patterns and mechanisms. We will then discuss the application, advantages and limitations of different antibiotic susceptibility testing methods that are currently in use based on the latest studies. We will also discuss the importance of antibiotic stewardship in the treatment of *H. pylori*, discuss how it can be achieved and compare the performances of empirical treatments with susceptibility-guided therapies in existing studies.

## 2. Antibiotic Resistance in *Helicobacter pylori*

Primary antibiotic resistance in *H. pylori* treatment can be defined as resistance in patients who have not started eradication therapy, whereas secondary antibiotic resistance occurs in patients who have previously undergone at least one unsuccessful eradication attempt [32]. Globally, primary and secondary resistance rates of clarithromycin, levofloxacin and metronidazole were all over 15% in all World Health Organization (WHO) regions [32]. The underlying mechanism behind antibiotic resistance by *H. pylori* is often due to genetic mutations that may inhibit the intracellular activation of antibiotics or change the drug target site altogether [33], and these mutations are mostly encoded chromosomally rather than in extrachromosomal elements [34,35,36]. The specifics of such genetic changes depend on the class of antibiotics (Figure 1) (Table 1). Apart from genetic sequence mutations, there are other possible mechanisms such as efflux pumps through the upregulation or downregulation of *hefABC* genes, or through biofilm formation, which can obstruct drug penetration [33] (Figure 1).

### 2.1. Resistance to Clarithromycin

Clarithromycin resistance in *H. pylori* is most commonly due to point mutations in domain V of 23S rRNA, especially A2143G, A2142G and A2142C [33,37,38,39], with A2143G causing the lowest eradication rate [56,57]. Other mutations have also been observed in [58,59,60] and outside of domain V [61], as well as in genes outside of 23S rRNA, such as *infB* and *rpl22*, which putatively affect 23S rRNA [40,41,42]. Increasing clarithromycin resistance was responsible for the failure of clarithromycin-containing eradication therapies [4,20,62]. The primary clarithromycin resistance rate was below 15% only in the WHO regions of the Americas and Southeast Asia, and when secondary resistance was considered, all WHO regions had >15% resistance rate [32], which has exceeded the suggested threshold at which clarithromycin triple therapy could be used as an empirical first-line treatment [63]. Studies found that primary clarithromycin resistance rates were 17–27% in Asia [64,65], 25% in treatment-naïve patients in Europe [66], 17.6–31.5% in the US [67,68] and 29.2% in Africa [69].

Of note, resistance caused by prior usage of clarithromycin or other macrolides for other diseases in patients who had never taken clarithromycin-based eradication therapy may be considered as secondary resistance [70,71]. Information on prior macrolide use, though difficult to obtain in real practice [72], should be collected whenever possible as cross resistance can occur within the same family of antibiotics [20]. Studies showed that macrolide exposure even in the past 10 to 14 years still correlated with *H. pylori* eradication failure [71,72,73,74,75] and was an independent risk factor for *H. pylori* clarithromycin resistance [71]. One such study found that those who previously took macrolides for >2 weeks had a significantly higher failure rate of eradication than those who took macrolides for ≤2 weeks (44.8% vs. 29.3%, *p* = 0.047), suggesting a duration effect of prior macrolide use on the success rate of eradication therapy [73].

### 2.2. Resistance to Metronidazole

The mutational changes involved in metronidazole resistance have a much more diverse repertoire and the genotype–phenotype correlation is often much more complex [4,33]. More well-known ones included *RdxA* and *FrxA* mutations, which were metronidazole-reducing enzymes and resulted in reduced drug activation [33]. Other putative mutations might result in increased efficiency of deoxyribonucleic acid (DNA) repair or oxidative stress response [33] and upregulated efflux [44,45]. It is possible that metronidazole resistance is caused by the cumulative effects of multiple pathways rather than a single mechanism, though until now, there has been no study that comprehensively investigated these mechanisms in the same clinical isolates [33].

The metronidazole resistance rate is generally much higher than that of clarithromycin, in part because of its widespread use to treat parasitic infections, urinary tract infections and gastrointestinal infections by anaerobes [76]. The primary and secondary resistance rates of metronidazole were over 15% in all WHO regions, with the highest rates of 56% and 65%, respectively, observed in the eastern Mediterranean region [32]. An increasing trend of metronidazole resistance has been observed in most of the WHO regions [32]. Studies have found that primary metronidazole resistance was 44% in the Asia-Pacific region [64], 30–40% in Europe [66,77], 42–43% in the US [67,68] and 48.7% in Africa [78]. A notable issue was the dual resistance to clarithromycin and metronidazole, which had a prevalence rate of around 8–15% in Europe [32,66,77], 6–11% in Asia [32] and 3–11% in the Americas [32,67,68]. Such dual resistance greatly reduces the efficacy of non-bismuth quadruple therapy [79,80]. In contrast to clarithromycin, in vitro metronidazole resistance can be overcome by increasing the dose, frequency and duration of therapy [63,81,82,83].

### 2.3. Resistance to Levofloxacin

As *H. pylori* does not naturally possess the genes for topoisomerase IV, levofloxacin resistance in *H. pylori* mainly arises from point mutations in the quinolone resistance-determining region (QRDR) of the *gyrA* or, to a lesser extent, *gyrB* gene, which code for DNA gyrase subunits A and B, respectively [33,46]. Mutations at *gyrA* mainly occur at codons N87 or D91 [84,85], while E463 mutation has been observed at *gyrB* [86,87].

Primary resistance against levofloxacin exceeded 15% in all WHO regions except Europe, while secondary resistance exceeded 15% in all regions with a significant rising trend observed in the western Pacific WHO region [32]. Other studies found that the primary levofloxacin resistance rates were 18% in the Asia-Pacific region [64], 15–20% in Europe [66,77], 37.6% in the US [68] and 17.4% in Africa [69]. Multiple resistance, commonly as dual or triple resistance to clarithromycin and/or metronidazole, is also an area of concern, with rates up to 6–10% in Europe [66,77].

### 2.4. Resistance to Amoxicillin

Resistance to amoxicillin in *H. pylori* are mainly caused by mutations in the *pbp1A* gene [47,48] that altered the penicillin-binding motifs SXXK, SXN and KTG, hence reducing the affinity of amoxicillin to PBP [33,37]. Enhancing effects can be contributed to by mutations in *pbp2* and *pbp3*, and if mutations in these three genes occur simultaneously, resistance could increase by over 200-fold [49]. Mutations of porin genes (e.g., *hopB* and *hopC*) or efflux pump-coding genes (e.g., *hefC*) can contribute to amoxicillin resistance in *H. pylori* as well [50,51].

Amoxicillin resistance in *H. pylori* is generally lower than other antibiotics. Primary and secondary amoxicillin resistance rates were previously found to be below 15% in all WHO regions [32]. Low resistance was generally found in more-developed regions, such as 6.4% in the US [67], 3% in the Asia-Pacific region [64] and 0.2% in Europe [77]. In Africa, the resistance rate could be as high as 72.6% [69], likely caused by the over abuse of amoxicillin because of low cost [88].

### 2.5. Resistance to Tetracycline

Tetracycline resistance in *H. pylori* is mainly caused by mutations at positions 926–928 of the 16S rRNA [52,53,54] that can lead to reduced drug affinity as they are located at the primary binding site [89]. Among these mutations, triple base-pair mutations (e.g., AGA → TTC) conferred a higher level of tetracycline resistance than single or double base-pair mutations [90]. Other possible mechanisms included efflux pump mechanisms that might be mediated by *HefABC* [55] or by proton motive force (PMF)-dependent mechanisms [91], which might explain resistance in *H. pylori* strains without 16S rRNA mutations.

Similar to amoxicillin, resistance against tetracycline is generally low around the world, with primary and secondary resistance below 10% in all WHO regions [32]. The overall resistance rate of tetracycline was 4% in the Asia-Pacific region [64], below 1% in Europe [66,77] and below 3% in the US [67,68]. In Africa, however, misuse of tetracyclines caused the resistance rate to be much higher at 49.8% [69], which might cause reduced efficacy of bismuth quadruple therapy locally [63].

## 3. Antibiotic Susceptibility Testing for *Helicobacter pylori*

Current antibiotic susceptibility testing (AST) methods for *H. pylori* are mainly divided into culture-based techniques and molecular-based methods. Figure 2 shows the comparison between these two main approaches in terms of their performance in studies, advantages and disadvantages.

### 3.1. Culture-Based Techniques

Traditionally, culture-based techniques, such as the agar dilution method, gradient diffusion susceptibility testing (E-test), broth microdilution method and disc diffusion method, are used as the gold standard for the AST of *H. pylori*, through which antibiotic susceptibility or resistance is inferred from the minimal inhibitory concentration (MIC) of the antibiotic being tested [99,100]. Broth microdilution and disc diffusion are not routinely used for *H. pylori* AST because of issues with standardization and accuracy in slow-growing bacteria such as *H. pylori* [33,101,102]. Agar dilution is a reliable method that can be adapted to test multiple *H. pylori* strains simultaneously and was recommended as a reference assay for evaluating the accuracy of other methods [100,103]. E-test allows quantification of the disc diffusion method and has good correlation with that of the agar dilution method for most antibiotics, except metronidazole because of the lack of anaerobic preincubation of plates used [101,102,104]. In the routine clinical setting, E-test also has the advantage of being cheaper and less time consuming than agar dilution [104].

Although culture-based methods currently serve as the gold standard for AST, they require gastric biopsy samples, which can only be obtained through invasive endoscopy procedures. Moreover, their reliability is greatly affected by the samples provided and the conditions under which they are performed, hence rendering the process costly, labor intensive and time consuming, providing results after one to two weeks at best [33,105]. Examples of such factors included the site of biopsy, the number of samples obtained, the quality of the sample (e.g., whether gut commensal microbiota were present), the time interval between sampling and culture and the transport conditions (e.g., temperature, air exposure, etc.) [105,106]. The culture of *H. pylori* itself is also challenging, as it requires a microaerophilic environment and an appropriate medium that would not interfere with the AST (e.g., redox variations in the medium may affect metronidazole susceptibility testing) [35,106]. Therefore, in routine clinical practice, there is only a 60–80% chance of successful isolation of the bacterium for testing [106], and even if results are obtained, its interpretation is also subjective and dependent on experiment conditions [107]. A recent systemic review found that bacterial growth failed in around 20% of the patients included [108]. It also found that while culture-tailored treatment offered a higher eradication rate than empirical treatment, it was only able to achieve the suggested optimal eradication rate of >90% [4,13] in culture-positive patients when it was adopted before first- and second-line regimens but not if it was done after two or more eradication failures [108], further highlighting the limitations of the culture-based approach to AST. It should be noted that the first- and second-line empirical regimens compared in this systemic review were mostly non-bismuth-based. A meta-analysis that included 54 studies found that a susceptibility-guided strategy had slightly better efficacy than an empirical clarithromycin-based triple therapy for first-line treatment but not for first-line quadruple empirical regimens (both with and without bismuth) [29].

### 3.2. Molecular-Based Techniques

Molecular-based techniques work by detecting specific mutations in *H. pylori* that encode resistance mechanisms and, in general, offer advantages such as a higher degree of standardization and reproducibility, a shorter time required for testing (often producing results within the same day) and the possibility of using specimens that can be obtained through non-invasive means and do not require immediate processing [105]. Molecular-based techniques can be applied to a variety of specimens, including fresh, frozen or paraffin-embedded formalin-fixed (FFPE) gastric biopsy samples, stool samples or gastric juice [33]. Current molecular-based methods are largely divided into PCR-based assays and next-generation sequencing (NGS) techniques, and their performances in different studies are summarized in Table 2.

(1)PCR-based assays

Most currently available assays are PCR-based assays that test for the most common point mutations in 23S rRNA for clarithromycin resistance and in *gyrA* for levofloxacin resistance [33,92,105]. In contrast, the development of assays for detecting metronidazole and amoxicillin resistance is challenging because of the complex underlying mechanisms of resistance [33]. In particular, assays for detecting clarithromycin resistance have been more well developed, and these assays could be applied to both biopsy and stool specimens and be used to detect *H. pylori* presence and clarithromycin resistance at the same time, with some even able to distinguish high or low levels of resistance based on the point mutation [95,109]. These real-time PCR assays also have a much faster turnaround time of around 2 to 6 h compared to culture-based techniques [105]. However, although stool sampling enabled the possibility of non-invasive testing, the sensitivity of molecular assays in detecting *H. pylori* using stool samples was not consistent in different studies, with one study finding a good sensitivity of 93.8% [23], while others found lower sensitivity (ranging from 63% to 84%) compared to traditional biopsy specimens [24,25,26]. The GenoType HelicoDR assay (Hain Life Science, Germany) has high sensitivity in detecting clarithromycin (ranging from 94–100%) and levofloxacin (ranging from 82.6–100%) when conducted on gastric biopsy samples [92,93,94], but one study found low concordance (clarithromycin: 52.9%; levofloxacin: 35.3%) of results obtained from stool samples with that from biopsy samples [97]. Such variance in sensitivity was likely dependent on the DNA extraction method and molecular assay used. Even if biopsy samples were used, the accuracy of such assays were also affected by the quality of the sample, as well as purity and condition of the DNA. False-negative results could arise if paraffin-embedded gastric biopsy samples were used as the DNA was broken into small pieces by the fixative [110,111,112]. The traditional Sanger sequencing method for identifying mutations after PCR also only has a limited coverage of nucleotides and cannot identify complex structural variants related to antibiotic resistance and may also not be cost effective if it is intended for use in routine settings because of its relatively higher price compared to conventional culture-based methods [33,92,110].

(2)Next-generation sequencing

Next-generation sequencing (NGS)-based methods have emerged as a potential alternative to both culture-based methods and current PCR-based assays, as they allow the identification of much more complex genetic variants that are involved in antibiotic resistance. NGS refers to methods that are developed after Sanger and Maxam–Gilbert sequencing and that allow massive parallel sequencing of DNA and RNA at a relatively low cost [113]. Whole-genome sequencing (WGS) is one of the applications of NGS technology in sequencing the entire genome of an organism and has enabled both the prediction of antibiotic resistance based on point mutations detected on target genes [35], as well as the detection of novel genetic mutations in clinical isolates. This is in stark contrast to conventional PCR-based assays that only detect known mutations that are most prevalent. The use of WGS to detect novel resistance-related mutations has not only been applied to clarithromycin and levofloxacin [36,43] but also to metronidazole [114] and amoxicillin [36,115]. This is particularly important for discovering new potential mechanisms that can explain resistance, as well as to provide new candidate genes that confer a better genotypic–phenotypic correlation than existing ones. A recent study conducted in Shenzhen, the southern part of China, detected a novel Gln31Arg mutation in *fliJ* in clarithromycin-resistant strains and levofloxacin-resistant strains, as well as a Ser176Thr/Ala mutation in *cheA* in levofloxacin-resistant strains, thus identifying *fliJ* and *cheA* as new candidate genes that might predict clarithromycin and levofloxacin resistance in *H. pylori* [43]. The same study also found no significant difference in the genotype of *gyrA* between *H. pylori* with phenotypic levofloxacin-resistance and those without, which was in concordance with a previous study finding inconsistent genotype-to-phenotype correlation of *gyrA* in predicting levofloxacin resistance [116], further highlighting the importance of continuous detection of novel mutations. Another recent study conducted in China detected the novel mutations N118K in *Fur* and Q242K in *Ribf*, as well as K219Q/N and H705fs in *Omp11* in metronidazole-resistant *H. pylori* strains [114]. Two studies found a total of seven new putative genotypes in *pbp1A* that might cause amoxicillin resistance [36,115]. A recent study conducted on 112 *H. pylori* strains from China even identified as many as 75, 5 and 13 new unique genes in metronidazole, clarithromycin and levofloxacin-resistant categories, respectively [117].

The accuracy of using NGS to predict antibiotic resistance was previously shown to be satisfactory and largely similar to that of culture-based methods, which are considered the gold standard for AST. Hulten et al. conducted paired comparisons between NGS and MICs obtained from culture using agar dilution for their determination of antibiotic resistance to clarithromycin, metronidazole, levofloxacin and amoxicillin [96]. When performed on clinical isolates, NGS showed good agreement with the agar dilution method for clarithromycin (k = 0.90012, *p* < 0.0001) and levofloxacin (k = 0.78161, *p* < 0.0001), while agreement with agar dilution for metronidazole (k = 0.5588, *p* < 0.0001) and amoxicillin (k = 0.21400, *p* = 0.0051) was less satisfactory. The accuracy of NGS in predicting resistance in clinical isolates was 97.1% for clarithromycin, 89.5% for levofloxacin, 77.6% for metronidazole and 95.9% for amoxicillin. Similarly, when performed on FFPE gastric biopsies, NGS showed good agreement with agar dilution for clarithromycin (k = 0.81236, *p* < 0.0001) and levofloxacin (k = 0.74953, *p* < 0.0001), with less satisfactory agreement for metronidazole (k = 0.54645, *p* < 0.0001) and amoxicillin (k = 0.21400, *p* = 0.0051). The accuracy of NGS in predicting resistance in FFPE biopsies was 94.1% for clarithromycin, 87.7% for levofloxacin, 77.1% for metronidazole and 95.9% for amoxicillin. Based on this, Moss et al. recently expanded further to investigate the accuracy of NGS in predicting antibiotic resistance in stool samples as well and found that the results obtained from stool samples was concordant with that from FFPE gastric biopsies in 91.4% of cases and that from fresh gastric specimens in 92.2% of cases [98]. The agreement between stool and fresh gastric samples was good for clarithromycin (k = 0.94), levofloxacin (k = 0.88) and metronidazole (k = 0.89). This was in stark contrast with conventional PCR assays, which might not be sensitive enough if performed on stool samples, hence highlighting NGS as a potentially reliable tool that can enable non-invasive means of AST. More studies on this subject, however, will have to be conducted to verify the performance of NGS on different clinical samples and in different geographical regions.

The use of NGS, in particular WGS, for AST in *H. pylori* strains faces several limitations. First, the accuracy of WGS when applied to gastric biopsies might be hampered by high human DNA background and low bacterial DNA content [36,118,119]; hence, suitable DNA extraction methods need to be carefully chosen and used. Second, as genotypic and phenotypic resistance may not always be correlated, further studies are required to verify the predictive accuracy of novel genotypes detected by NGS using phenotypic outcomes or clinical observations, or through retrospective analysis of the sequencing data [33,120,121,122]. Studies are also required to assess the relative importance of each novel mutation in causing resistance, which is important for routine clinical explanations. The explanation of novel genotypes in causing resistance will in part rely on an existing understanding of the molecular basis of causing resistance, and new mechanisms will need to be verified in follow-up studies, such as through knock-out studies.

In addition, current studies that have assessed the accuracy of NGS or WGS in predicting antibiotic resistance are limited by small sample sizes for certain antibiotics (e.g., tetracycline and rifabutin) because of low prevalence of antibiotic resistance [35,96]. Hence, studies with wider gene coverages, larger sample sizes or multicenter design from different geographical regions are warranted. Standardized and user-friendly computational software and tools need to be developed so that NGS data can be easily analyzed and applied in routine clinical settings [33,36]. NGS databases also require regular and continuous updating of novel mutations detected; otherwise, underestimation of resistance mechanisms may occur because of them not being represented in the databases [122]. Despite these limitations, advancements in sequencing technology and increasing knowledge of molecular mechanisms of resistance will undoubtedly boost the role of NGS as a fast and reliable tool for AST in the context of *H. pylori* infection in the future.

## 4. Antibiotic Stewardship in the Treatment of *Helicobacter pylori* Infection

### 4.1. Issues of the “Better-than Approach” in H. pylori Treatment and Research

Subtherapeutic levels of an antibiotic in *H. pylori* treatment is the main culprit for the development of resistance against it, as it favors the survival and natural selection of these resistant strains [33]. Prior antibiotic usage within 180 days was found to be associated with higher risk of needing retreatment for *H. pylori* [123]. Antibiotic stewardship has been increasingly promoted as an effective solution to this growing problem. In essence, antibiotic stewardship refers to sets of actions that (1) promote and monitor the responsible use of antibiotics, (2) promote the selection of optimal treatment regimens and (3) ensure the sustainable access to antibiotics by those in need [13,124,125]. This is especially important in *H. pylori* treatment as antibiotic combinations in treating *H. pylori* have long been based on trial and error, and treatment regimens were chosen because they were relatively “better than” traditional empirical regimens but not because they had an acceptable absolute cure rate [13,125]. The adaptation of antibiotic stewardship principles in *H. pylori* treatment will mean that the effectiveness of treatment should instead be determined as whether the treatment can achieve a prespecified success rate in both routine clinical settings and in clinical trials.

The use of the “better-than” approach in clinical trials had several issues. On one hand, its value might be undermined if at least one or even all of the treatment arms tested failed to achieve the optimal success rate stipulated by antibiotic stewardship [125]. On the other, ethical concerns might arise as participants were often not informed of any possible suboptimal eradication rate, which was actually predictable based on prior experience [125,126,127]. Similarly, overfocus on relative comparisons in meta-analysis means that the results may be misguided by inherent variations in the clinical trials included, such as geographical variations of resistance patterns [125]. With antibiotic stewardship, only therapies with reliable high cure rates will remain of interest to physicians, and comparisons will only be made among them [13].

### 4.2. Susceptibility-Guided Therapy vs. Empirical Therapy in Achieving Antibiotic Stewardship

Antibiotic stewardship also entails the principle that only antibiotics to which the bacteria are susceptible should be used in treatment, and ideally, only directly (based on AST) or indirectly (based on test-of-cure results) susceptibility-guided therapy (SGT) optimized for the population being treated should be used [13]. However, empirical therapies used by physicians for treating *H. pylori* are not often based on the latest information of local antibiotic resistance patterns because such data are either not available or not shared with physicians systematically. When optimized, empirical therapy can be effective in *H. pylori* treatment. The results of five meta-analyses found that SGT was superior to empirical regimens in first-line treatment [22,27,28,29,128]. However, the studies included were highly heterogeneous, and there might be no significant difference when each empirical regimen was analyzed separately (Table 3). The most recent meta-analysis, which included the most studies, found that SGT (using either culture or PCR) was superior to empirical clarithromycin-containing triple therapy for first-line treatment in high (>20%) clarithromycin resistance areas (RR: 1.13, 95% CI: 1.03–1.25) and also in low clarithromycin resistance areas (RR: 1.24, 95% CI: 1.15, 1.32) [29]. However, there was no significant difference in eradication rates between SGT and first-line quadruple therapy [29]. Paradoxically, there was also no significant difference between SGT and empirical regimens in rescue treatments, similar to two other meta-analysis [27,28,29]. This interesting phenomenon might be due to the paucity and high heterogeneity of the studies included [128] or because the efficacy of rescue SGT could have been affected by a previous first-line regimen choice [28]. The lack of evidence in the use of SGT in rescue therapies was acknowledged by the Maastricht VI report, though it recommended the routine monitoring of antibiotic resistance patterns by AST or eradication rate data to optimize empirical rescue therapies [4].

In recent years, there have been more studies that compared PCR-based SGT to empirical regimens. A study in France found that PCR-based SGT had a significantly higher eradication rate than empirical triple therapy [129]. Similar findings were seen in a Taiwan study with PCR conducted on gastric juice samples [130] and several Korean studies that used dual priming oligonucleotide (DPO)-based PCR on gastric biopsies [131,132]. However, other Korean studies found no significant differences when DPO-PCR-based SGT was compared to quadruple therapies, both bismuth- [30,31,133] or non-bismuth-containing [134,135]. A Taiwan study also found no significant difference in eradication rates of PCR-based SGT and empirical therapy based on medication history in patients with ≥2 failed eradication attempts [136]. Another Taiwan study showed that SGT based on PCR on gastric biopsies could achieve a similar eradication rate as culture-based SGT in first-line treatment (86% vs. 87%, *p* = 0.81) and in third-line treatment (88% vs. 87%, *p* = 0.74) [137], hence supporting the use of PCR methods to substitute culture as the method of AST if SGT strategy is to be used.

Alternatively, it was recently shown that a treatment regimen derived from routine evaluation of mutations associated with antibiotic resistance using NGS on gastric biopsy samples offered 4.4-greater odds of eradication than using empirical regimens [138]. That being said, comparative studies of NGS on other samples, such as stool, are limited. More studies are also needed to compare molecular-testing-guided treatment to empirical bismuth quadruple treatment (BQT), which can achieve approximately 90% eradication rates in the Western population [139].

Apart from efficacy, the cost effectiveness of the susceptibility-guided strategy is another factor that must be taken into consideration. However, as in the case of efficacy, studies that have attempted to compare the cost effectiveness of a susceptibility-guided strategy with empirical treatment did not find consistent conclusions [106]. The continuous improvement of molecular-based methods may be able to offer a lower cost than culture-based testing and raise the cost effectiveness of susceptibility-guided strategy in the future, though further studies are required to provide clinical evidence [4,140]. Current evidence suggests that PCR-based SGT might not be more cost effective than an up-to-date empirical treatment, such as BQT. Two Korean studies found that DPO-PCR-based SGT would offer lower costs than empirical triple therapy only when the latter’s eradication rate dropped to below 75.3% to [132] 80% [131]. In contrast, one Korean study found that the total medical cost was lower for SGT compared to empirical treatment (including triple therapy, sequential therapy and BQT) because of significantly higher treatment costs in the empirical group [141]. When only BQT was under comparison, two Korean studies found the empirical BQT was more cost effective than DPO-PCR-based SGT [30,31]. In these studies, the cost of PCR-based SGT ranged from USD 90.3 [31] to USD 503.5 [30]. A Taiwan study also found that compared to empirical treatment based on medication history, PCR-based SGT required USD 6920 to additionally cure one patient with refractory *H. pylori* infection, which undermined SGT’s cost effectiveness. The use of NGS for designing SGT in routine clinical practice is also hindered by the need of specialized machines, hardware and software for handling large genomic data, as well as trained staff for operation, which together made NGS very expensive (reaching USD 2000–3000 per sample merely for pathogen detection) and not suitable for routine use in the short term [142].

Therefore, empirical regimens, if based on latest local antibiotic susceptibility patterns, are likely more cost effective than SGT. The performance of empirical therapies should be regularly monitored so that they can quickly adapt to changes in antibiotic susceptibility patterns. In recent years, using vonoprazan, a potassium-competitive acid blocker with a higher acid suppressive potency than PPIs, in eradication treatment has been shown to achieve a >90% eradication rate in several clinical trials in Asia but not in the West (Table 4) [143,144,145,146,147,148,149,150]. Whether empirical use of a vonoprazan-based regimen is non-inferior to a susceptibility-guided strategy in terms of *H. pylori* eradication warrants further investigation.

### 4.3. Measures to Achieve Antibiotic Stewardship in H. pylori Treatment

In the meantime, analyzing routinely collected clinical records of diagnosis, treatment and the test of a cure (such as urea breath test) conducted at least four weeks after the treatment is completed [13,125,157] is an existing method that can indirectly assess antibiotic susceptibility. Yet, despite the expected lower cost and relative ease in implementation, these routinely collected data are often not shared with physicians to guide their clinical decisions or not included in antibiotic surveillance programs so that they can be analyzed and incorporated into treatment guidelines systematically [13,125]. The latest Maastricht VI Florence consensus report stated that “it is reasonable to recommend that susceptibility tests (molecular or after culture) are routinely performed, even before prescribing first-line treatment, in respect to antibiotic stewardship” [4]. As molecular methods for AST become more convenient and affordable, the long-term goal will be to adapt an individualized approach in the design of treatment regimens, so that the highest eradication rate can be achieved while avoiding the use of unnecessary antibiotics (i.e., antibiotics that in fact do not contribute to treatment effectiveness) in treatment combinations to combat global resistance [13,125,158,159].

The implementation of antibiotic stewardship in *H. pylori* treatment requires collective effort from the individual, local and up to international level. At the individual level, patient compliance is an important factor deciding treatment effectiveness. A study found a global odds ratio (OR) of around 7 in its association with treatment effectiveness, and that excellent compliance was the factor that was mostly associated with eradication rates in all treatment regimens evaluated [12]. Another meta-analysis also identified that patient compliance, with an OR of 16, was the most significant factor affecting treatment success [160]. Therefore, the best way the individual patient can help in achieving antibiotic stewardship is to comply with the treatment regimen prescribed by the physician. This must be emphasized to the patient by the physician during consultation or through public education campaigns [161,162]. Physicians must also be educated on up-to-date antibiotic resistance patterns and relevant guidelines concerning antibiotic use [161,162]. Local healthcare authorities should take the initiative to establish antibiotic stewardship programs that provide AST in routine clinical setting, monitor antibiotic prescriptions and resistance patterns and provide treatment guidelines tailored to local conditions [125,161,162]. On the international level, consensus reports and guidelines play an important role in promoting antibiotic stewardship measures and providing treatment suggestions based on updated antibiotic resistance pattern, but this potential has not been fully reached up until now [125]. Existing antibiotic surveillance programs, both at local and international level, should include *H. pylori* if it has not already been done [125]. Collaborative efforts from all sorts of stakeholders are thus crucial in ensuring that antibiotic stewardship is applied to *H. pylori* treatment around the globe.

## 5. Conclusions

Antibiotic stewardship should be practiced in view of increasing antibiotic resistance of *H. pylori* worldwide to ensure the sustainability of current treatment regimens. Antibiotic susceptibility testing becomes increasingly important to provide data on antibiotic resistance patterns to guide antibiotic stewardship measures. With advancements in technology, molecular-based sequencing techniques allow the detection of antibiotic resistance in a rapid and convenient manner. In particular, preliminary evidence showed that stool-based molecular tests have potential to serve as a non-invasive and cost-effective AST tool in routine clinical practice, though more studies are needed to establish its accuracy and costs for it to be of use in the future. Novel genetic mutations may also be discovered with WGS, although validation studies are required. Collaboration at the individual, local and international levels is needed to implement various measures aimed at improving treatment effectiveness and patient compliance that are of utmost importance for the application of antibiotic stewardship in *H. pylori* treatment.

## Figures and Tables

**Figure 1 ijms-24-11708-f001:**
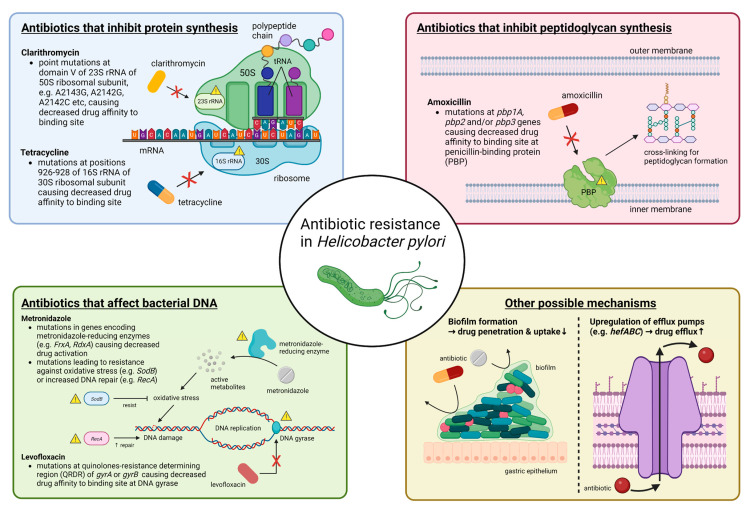
Summary of molecular mechanisms of antibiotic resistance in *H. pylori*. In general, antibiotic resistance in *H. pylori* arises because of mutations that decrease drug affinity to its binding site. Examples include mutations in 23S rRNA for clarithromycin, in *pbp1A* for amoxicillin, in *gyrA* or *gyrB* for levofloxacin and in 16S rRNA for tetracycline. Metronidazole is unique in that it is a prodrug and works by producing active metabolites that can damage bacterial DNA after reductive activation. Mutations causing metronidazole resistance have a much more diverse repertoire, and more well-known ones include those in genes encoding for metronidazole-reducing enzymes (e.g., *FrxA* and *RdxA*) causing decreased drug activation, as well as those that lead to resistance to oxidative stress caused by metronidazole or increased DNA repair. Other possible mechanisms that may apply to most antibiotics include increased drug efflux caused by upregulation of efflux pump genes (e.g., *hefABC*) or biofilm formation, which decrease drug penetration. Abbreviations: rRNA, ribosomal RNA; mRNA, messenger RNA; tRNA, transfer RNA; DNA, deoxyribonucleic acid.

**Figure 2 ijms-24-11708-f002:**
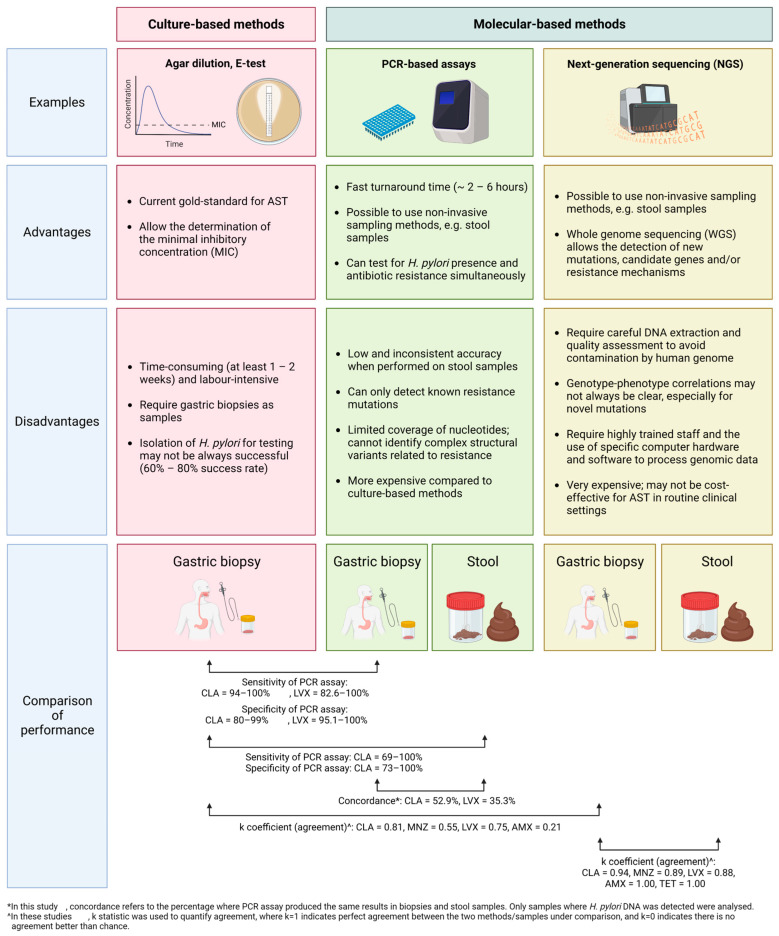
Comparison of culture-based and molecular-based antibiotic susceptibility testing (AST) methods. Culture-based methods, such as agar dilution or E-test, are the current gold standard for AST in *H. pylori*. However, they are time consuming and labor intensive and require invasive procedures to obtain gastric biopsy samples for testing. The isolation of *H. pylori* for testing is also challenging. In comparison, molecular-based methods, such as PCR-based assays, can be applied to other samples, such as stool samples, that can be obtained through non-invasive means. However, the performance of PCR-based assays on stool samples was inconsistent and requires further investigation. Also, PCR-based assays can only detect known mutations of resistance. NGS is a relatively newer technique and can be used to detect novel mutations or resistance mechanisms by means of WGS. However, NGS is very expensive and is difficult to incorporate into routine clinical use in the short term. Studies have attempted to compare the accuracy of molecular-based methods to culture-based methods with consideration of the sample types used. In general, there was good concordance between culture- and molecular-based methods when gastric biopsies were used, though results varied in different antibiotics [92,93,94,95,96]. The concordance was poorer when stool samples were used [25,26,95,97], though recent studies had shown preliminary evidence that NGS might provide better accuracy in stool samples with good concordance to results achieved on gastric biopsies [98]. Abbreviations: PCR, polymerase chain reaction; NGS, next-generation sequencing; AST, antibiotic susceptibility testing; WGS, whole-genome sequencing; CLA, clarithromycin; LVX, levofloxacin; MNZ, metronidazole; AMX, amoxicillin; TET, tetracycline.

**Table 1 ijms-24-11708-t001:** Molecular basis of resistance against antibiotics commonly used in *Helicobacter pylori* eradication therapies.

	Genes Involved	Resistance Mechanisms
Clarithromycin	Domain V of 23S rRNA (most prevalent: A2143G, A2142G, A2142C) [33,37,38,39]	Decrease drug affinity to its binding site
	*infB* (translation initiation factor IF-2), *Rpl22* (Ribosomal protein L22) [40,41,42]	Putative effects on 23S rRNA (the drug target) *
	**Novel gene candidate:** *fliJ* (flagellar export protein): Gln31Arg [43]	Putative effects on bacterial chemotaxis and flagellar motility *
Metronidazole	*RdxA* (oxygen-insensitive NAD(P)H nitroreductase), *FrxA* (NAD(P)H flavin nitroreductase),	Mutations in metronidazole-reducing enzymes coding genes causing decreased drug activation
	*FdxA* (ferredoxin), *FdxB* (ferredoxin-like protein), *FldA* (flavodoxin) [33]	Mutations in putative metronidazole-reducing enzymes coding genes causing decreased drug activation
	*SodB* (superoxide dismutase), *Fur* (ferric uptake regulator), *MdaB* (modulator of drug activity B) [33]	Resistance against oxidative stress brought by metronidazole
	*RecA* [33]	Upregulation of *RecA* causing enhanced DNA repair against damages brought by metronidazole
	*Ribf* (riboflavin biosynthesis protein), *Omp11* [33]	Putative association with metronidazole resistance *
	*hefA* (efflux pump) [44,45]	Increased drug efflux causing below lethal intracellular concentrations
Levofloxacin	*gyrA* (DNA gyrase subunit A): mainly at codons N87 or D91, *gyrB* (DNA gyrase subunit B) [33,46]	Mutations in quinolones resistance-determining region (QRDR) causing decreased drug affinity to its binding site
	**Novel gene candidates:** *fliJ* (flagellar export protein): Gln31Arg [43] *cheA* (histidine kinase): Ser176Thr/Ala [43]	Putative effects on bacterial chemotaxis and flagellar motility *
Amoxicillin	*pbp1A* (penicillin-binding protein 1A) [47,48] *pbp2* (penicillin-binding protein 2) [49] *pbp3* (penicillin-binding protein 3) [49]	Mutations altering penicillin-binding motifs SXXK, SXN and KTG, causing decreased drug affinity to its binding site
	*hopB, hopC* (porins), *hefC* (efflux pump) [50,51]	Increased drug efflux causing below lethal intracellular concentrations
Tetracycline	16S rRNA positions 926–928 [52,53,54]	Decrease drug affinity to its binding site
	*hefABC* (efflux pump) [55]	Increased drug efflux causing below lethal intracellular concentrations

* Hypothetical mechanism.

**Table 2 ijms-24-11708-t002:** Summary of studies assessing the performance of polymerase chain-reaction-based assays and next-generation sequencing in predicting *H. pylori* antibiotic resistance.

Study	Sample Size and Specimen Samples	Methods Tested	Antibiotics Tested	Results
**PCR-based assays**				
Fernandez-Caso et al., 2022 [92]	223 gastric biopsy samples studied for antibiotic susceptibility	GenoType HelicoDR assay (Hain Life Science, Nehren, Germany) Control: E-test	Clarithromycin	Sensitivity: 99.1% Specificity: 80.0%
			Levofloxacin	Sensitivity: 100% Specificity: 100%
Cambau et al., 2009 [93]	92 clinical strains and 105 gastric biopsy samples	GenoType HelicoDR assay (Hain Life Science, Germany) Control: E-test	Clarithromycin	Sensitivity: 94% Specificity: 99% Concordance score: 0.96
			Levofloxacin	Sensitivity: 87% Specificity: 98.5% Concordance score: 0.94
Deyi et al., 2011 [94]	64 biopsy samples compared	GenoType HelicoDR assay (Hain Life Science, Germany) Control: E-test	Clarithromycin	Sensitivity: 100% Specificity: 86.2% PPV: 89.7% NPV: 100%
			Levofloxacin	Sensitivity: 82.6% Specificity: 95.1% PPV: 90.5% NPV: 90.7%
Brennan et al., 2016 [97]	55 biopsy specimens and 66 stool specimens compared	GenoType HelicoDR assay (Hain Life Science, Germany) on stool specimens Control: GenoType HelicoDR assay on biopsy specimens	Clarithromycin	Concordance: 52.9%
			Levofloxacin	Concordance: 35.3%
Redondo et al., 2018 [109]	60 gastric biopsies	Lightmix RT-PCR (TIB Molbiol, Berlin, Germany) Control: E-test	Clarithromycin	95% concordance rate between RT-PCR and E-test
Schaberiter-Gurtner et al., 2004 [95]	45 isolates (biopsy and stool samples) studied for antibiotic susceptibility	TaqMan RT-PCR (Meridian Bioscience, Newtown, OH, United States) Control: E-test	Clarithromycin	Sensitivity: 100% (biopsy), 100% (stool) Specificity: 82% (biopsy), 73% (stool) PPV: 100% (biopsy), 100% (stool) NPV: 94% (biopsy), 92% (stool)
Beckman et al., 2017 [23]	Total of 294 stool samples	TaqMan RT-PCR (Meridian Bioscience, United States) Reference: Treatment outcome as tested by UBT	Clarithromycin	Sensitivity in detecting *H. pylori* DNA: 93.8% Correlation between genotype prediction by PCR and eradication of infection: 86%
Lottspeich et al., 2007 [24]	Total of 100 stool samples from children	ClariRes RT-PCR (Ingenetix, Vienna, Austria) Control: E-test	Clarithromycin	For *H. pylori* detection: Sensitivity: 63% Specificity: 100% PPV: 100% NPV: 76.1% For clarithromycin susceptibility testing: 2 false negative results out of 6 clarithromycin resistant cases
Vecsei et al., 2009 [25]	Total of 143 stool samples from children	ClariRes RT-PCR (Ingenetix, Vienna, Austria) Control: E-test	Clarithromycin	For *H. pylori* detection: Sensitivity: 83.8% Specificity: 98.4% PPV: 98.5% NPV: 82.7% For clarithromycin susceptibility testing: Sensitivity: 89.2% Specificity: 100% PPV: 100% NPV: 88.2%
Scaletsky et al., 2011 [26]	Total of 217 stool samples from children	ClariRes RT-PCR (Ingenetix, Vienna, Austria) Control: E-test	Clarithromycin	For *H. pylori* detection: Sensitivity: 69% Specificity: 100% Test accuracy: 93.9% For clarithromycin susceptibility testing: Sensitivity: 83.3% Specificity: 100% Test accuracy: 95.6%
**Next-Generation Sequencing**				
Hulten et al., 2021 [96]	170 *H. pylori* clinical isolates and FFPE gastric biopsies for amoxicillin, clarithromycin and metronidazole 57 *H. pylori* clinical isolates and FFPE gastric biopsies for levofloxacin and tetracycline	PyloriAR NGS (American Molecular Laboratories, Vernon Hills, IL, United States) Control: Agar dilution	Clarithromycin	Sensitivity: 93.3% Specificity: 94.3% PPV: 77.8% NPV: 98.5% Accuracy: 94.1% Agreement (k coefficient): 0.81236 (*p* < 0.0001)
			Metronidazole	Sensitivity: 70.1% Specificity: 86.3% PPV: 87.2% NPV: 68.5% Accuracy: 77.1% Agreement (k coefficient): 0.54645 (*p* < 0.0001)
			Levofloxacin	Sensitivity: 87.9% Specificity: 87.5% PPV: 90.6% NPV: 84.0% Accuracy: 87.7% Agreement (k coefficient): 0.74953 (*p* < 0.0001)
			Amoxicillin	Sensitivity: 12.5% Specificity: 100% PPV: 100% NPV: 95.9% Accuracy: 95.9% Agreement (k coefficient): 0.21400 (*p* = 0.0051)
			Tetracycline	Only 1 out of 57 samples tested resistant to tetracycline by agar dilution; none of the 57 samples were tested tetracycline resistant by PyloriAR NGS
Moss et al., 2022 [98]	64 fresh gastric biopsy samples and stool samples compared for antibiotic susceptibility prediction	PyloriAR NGS (American Molecular Laboratories, United States) on stool samples Control: PyloriAR NGS on gastric biopsy samples	Clarithromycin	Agreement (k coefficient): 0.94 (95% CI: 0.90–1.00)
			Metronidazole	Agreement (k coefficient): 0.89 (95% CI: 0.76–1.00)
			Levofloxacin	Agreement (k coefficient): 0.88 (95% CI: 0.75–1.00)
			Amoxicillin	Agreement (k coefficient): 1.00
			Tetracycline	Agreement (k coefficient): 1.00

Abbreviations: RT-PCR, real-time polymerase chain reaction; NGS, next-generation sequencing; FFPE, formalin-fixed paraffin-embedded.

**Table 3 ijms-24-11708-t003:** Summary of meta-analysis comparing eradication rates of susceptibility-guided therapy with empirical therapy.

Meta-Analysis	No. of Studies Included in Meta-Analysis	Results
Wenzhen et al., 2010 [22]	5 RCTs (*I*^2^ = 0%)	**First-line treatment:** Superiority of SGT (Method of AST: culture-based; 2 antibiotics based on AST + PPI or Bismuth) over empirical triple therapy: RR: 1.19, 95% CI: 1.11–1.30, *p* < 0.001
Lopez-Gongora et al., 2015 [27]	Total: 8 RCTs, 4 quasi-RCTs First-line treatment: 5 RCTs, 4 quasi-RCTs; mild to moderate heterogeneity (*I*^2^ = 23–54%) Second-line treatment: 4 RCTs; high heterogeneity (*I*^2^ = 87%)	**First-line treatment:** Superiority of SGT (methods including culture-based methods and fecal PCR) to empirical treatment Including all studies: RR 1.16, 95% CI 1.10–1.23, *p* < 0.001 Including RCTs only: RR 1.15, 95% CI 1.07–1.24, *p* < 0.001 Including studies using empirical triple therapy only: RR 1.18, 95% CI 1.11–1.26, *p* < 0.001 **Second-line treatment:** No significant difference between SGT (culture-based methods) and empirical treatment (none are bismuth quadruple therapy): RR 1.11, 95% CI: 0.86–1.50, *p* = 0.38
Chen et al., 2016 [28]	Total: 10 RCTs, 3 non-randomized controlled clinical trials; high pooled heterogeneity (*I*^2^ = 57.1%) First-line treatment: 10 studies Rescue treatment: 4 studies Method of AST: ○ Culture-based: 10 studies ○ PCR: 3 studies Empirical regimen: ○ Standard triple therapy: 7 studies ▪ 7-day duration: 5 studies ▪ 10-day duration: 2 studies ○ Bismuth quadruple therapy: 3 studies ○ Sequential therapy: 2 studies ○ 14-day moxifloxacin-containing triple therapy: 1 study	**Pooled analysis:** Superiority of SGT to empirical treatment: RR 1.16, 95% CI 1.11–1.22 **First-line treatment:** Superiority of SGT to empirical treatment: pooled RR 1.18, 95% CI 1.14–1.22 **Rescue treatment:** No significant difference between SGT and empirical treatment: pooled RR 1.16, 95% CI 0.96–1.39 **Subgroup analysis:** ***Method of AST:*** Superiority of SGT to empirical treatment for both culture-based and PCR ○ Culture-based: pooled RR 1.14, 95% CI 1.08–1.21 ○ PCR: pooled RR 1.23, 95% CI 1.11–1.35 ***Empirical regimens:*** Superiority of SGT to: 7-day standard triple therapy (pooled RR 1.22, 95% CI 1.16–1.29) Bismuth quadruple therapy (pooled RR 1.15, 95% CI 1.08–1.22) 14-day moxifloxacin-containing triple therapy (pooled RR 1.27, 95% CI 1.08–1.51) No significant difference between SGT and: 10-day standard triple therapy (pooled RR 1.03, 95% CI 0.76–1.41) Sequential therapy (pooled RR 1.01, 95% CI 0.79–1.30)
Gingold-Belfer et al., 2021 [128]	Total: 16 RCTs (*I*^2^ = 75%) First-line treatment: 13 studies; high heterogeneity (*I*^2^ = 75%) Second-line treatment: 3 studies; high heterogeneity (*I*^2^ = 84%), hence meta-analysis not done Method of AST: ○ Culture-based: 14 studies ○ PCR: 5 studies	**Pooled analysis:** Superiority of SGT over empirical therapy: RR 1.14, 95% CI: 1.07–1.21, *p* < 0.0001 **First-line treatment:** Superiority of SGT over empirical therapy: RR 1.14, 95% CI: 1.07–1.21, *p* < 0.001 ***Subgroup analysis:*** *Empirical regimen:* ○ Superiority of SGT to empirical triple therapy: RR 1.19, 95% CI: 1.13–1.26, *p* < 0.001 ○ No significant difference between SGT and empirical quadruple therapy *Clarithromycin resistance:* ○ No significant difference between SGT and clarithromycin triple therapy when the prevalence of clarithromycin resistance is <20% (RR: 1.10, 95% CI 0.95–1.28, *p* = 0.199) ○ Superiority of SGT over clarithromycin triple therapy when the prevalence of clarithromycin resistance is >20% (RR: 1.18, 95% CI 1.07–1.30, *p* = 0.001)
Nyssen et al., 2022 [29]	All studies: Total: 31 RCT comparisons, 23 non-RCTs; high heterogeneity (*I*^2^ = 83%) First-line treatment: 30 studies; high heterogeneity (*I*^2^ = 83%) Rescue treatment (more than one treatment failure): 16 studies (*I*^2^ = 78%) Method of AST: ○ Culture-based: 36 studies (*I*^2^ = 83%) ○ PCR: 16 studies (*I*^2^ = 84%) Empirical regimen: ○ First-line quadruple therapy (both with and without bismuth): 12 studies (*I*^2^ = 72%) RCTs only: Total: 31 RCT comparisons (from 27 RCT studies) *; high heterogeneity (*I*^2^ = 74%) First-line treatment: 21 comparisons; high heterogeneity (*I*^2^ = 75%) Rescue treatment (more than one treatment failure): 8 comparisons (*I*^2^ = 84%) Method of AST: ○ Culture-based: 24 RCTs (*I*^2^ = 65%) ○ PCR: 8 RCTs (*I*^2^ = 85%) Empirical regimen: ○ First-line quadruple therapy (both with and without bismuth): 8 studies (*I*^2^ = 77%)	**A.** **All studies** **Pooled analysis:** Superiority of SGT to empirical treatment: RR 1.12, 95% CI: 1.08–1.17 **First-line treatment:** Superiority of SGT to empirical treatment: RR 1.13, 95% CI: 1.08–1.17 **Rescue treatment:** No significant difference between SGT and empirical treatment: RR 1.07, 95% CI: 0.97–1.18 **Subgroup analysis:** ***Method of AST:*** Superiority of SGT to empirical treatment for both culture-based and PCR ○ Culture-based: pooled RR 1.11, 95% CI 1.05–1.16 ○ PCR: pooled RR 1.08, 95% CI 1.01–1.16 ***Empirical regimen:*** ○ Superiority of SGT to first-line clarithromycin triple therapy in areas with: ▪ High (>20%) clarithromycin resistance: RR 1.13, 95% CI: 1.08–1.17 ▪ Low clarithromycin resistance: RR 1.24, 95% CI: 1.15–1.32 ○ No significant difference between SGT and first-line quadruple therapy: RR 1.04, 95% CI: 0.99–1.09 **B.** **RCTs only** **Pooled analysis:** Superiority of SGT to empirical treatment: RR 1.13, 95% CI: 1.07–1.18 **First-line treatment:** Superiority of SGT to empirical treatment: RR 1.14, 95% CI: 1.08–1.20 **Rescue treatment:** No significant difference between SGT and empirical treatment: RR 1.10, 95% CI: 0.85–1.42 **Subgroup analysis:** ***Method of AST:*** ○ Superiority of SGT to empirical treatment for culture-based methods in: ▪ Pooled analysis: RR 1.13, 95% CI 1.08–1.19 ▪ First-line treatment: RR 1.13, 95% CI: 1.07–1.20 ○ No significant difference of SGT to empirical treatment for: ▪ Culture-based methods in second-line treatment: RR 1.05, 95% CI: 0.95–1.17 ▪ PCR: pooled RR 1.10, 95% CI 0.99–1.24 ***Empirical regimen:*** ○ No significant difference between SGT and first-line quadruple therapy: RR 1.05, 95% CI: 0.99–1.12

* Different treatment groups in same RCT are meta-analyzed separately. Abbreviation: RCT, randomized clinical trials; RR, risk ratio; CI, confidence interval; SGT, susceptibility testing-guided therapy; AST, antibiotic susceptibility testing; PCR, polymerase chain reaction.

**Table 4 ijms-24-11708-t004:** Summary of randomized clinical trials comparing vonoprazan-based therapies to empirical eradication therapies (arranged in decreasing order of eradication rate by vonoprazan-based therapies).

Study	Country	Sample Size	Study Design	Special Patient Characteristics	Treatment Regimen	Results (Eradication Rates of *H. pylori*)	
						ITT	PP
**A.** **Vonoprazan dual or triple therapy vs. PPI triple therapy**							
* In treatment-naive patients *							
Bunchorntavakul et al., 2021 [143]	Thailand	122	Open-label Single-center	/	20 mg VPZ twice daily + 1 g AMX twice daily + 500 mg CLA twice daily for 7 days 20 mg OPZ twice daily + 1 g AMX twice daily + 500 mg CLA twice daily for 14 days	VPZ triple: 96.7% PPI triple: 88.5% *p* = 0.083	VPZ triple: 98.3% PPI triple: 93.1% *p* = 0.159
Maruyama et al., 2017 [145]	Japan	141	Single-blind Single-center	/	20 mg VPZ twice daily + 750 mg AMX twice daily + 200 mg or 400 mg CLA twice daily for 7 days; or 20 mg RPZ or 30 mg LPZ twice daily + 750 mg AMX twice daily + 200 mg (if RPZ) or 400 mg (if RPZ or LPZ) CLA twice daily for 7 days	VPZ-triple: 95.8% PPI-triple: 69.6% *p* < 0.001	VPZ-triple: 95.7% PPI-triple: 71.4% *p* < 0.001
Murakami et al., 2016 [146]	Japan	650	Double-blind Multi-center Noninferiority	All patients had history of gastric or duodenal ulcer	20 mg VPZ twice daily + 750 mg AMX twice daily + 200 mg or 400 mg CLA twice daily for 7 days 30 mg LPZ twice daily + 750 mg AMX twice daily + 200 mg or 400 mg CLA twice daily for 7 days	VPZ triple (regardless of CLA dose): 92.6% LPZ triple (regardless of CLA dose): 75.9% noninferiority *p* < 0.0001	NA
Ang et al., 2022 [151]	Singapore	252	Open-label Single-center Noninferiority	/	20 mg VPZ twice daily + 1 g AMX twice daily + 500 mg CLA twice daily for 7 days 20 mg OPZ or 20 mg EPZ or 20 mg RPZ twice daily + 1 g AMX twice daily + 500 mg CLA twice daily for 14 days	VPZ triple: 87.4% PPI triple: 88.0% noninferiority *p* < 0.05	VPZ triple: 96.3% PPI triple: 94.0% noninferiority *p* < 0.05
Chey et al., 2022 [152]	USA, Europe	1046	Open-label or double-blind dependent on treatment group Multi-center Noninferiority	All patients were not resistant to CLA or AMX	Open-label VPZ dual therapy (20 mg VPZ twice daily + 1 g AMX 3 times daily) for 14 days Double-blind triple therapy twice a day (20 mg VPZ twice daily or 30 mg LPZ twice daily + 1 g AMX twice daily + 500 g CLA twice daily) for 14 days	VPZ triple: 84.7% VPZ dual: 78.5% LPZ triple: 78.8% VPZ triple vs. LPZ triple: noninferiority *p* < 0.001 VPZ dual vs. LPZ triple: noninferiority *p* < 0.05	VPZ triple: 90.4% VPZ dual: 81.2% LPZ triple: 82.1% VPZ triple vs. LPZ triple: noninferiority *p* < 0.001 VPZ dual vs. LPZ triple: noninferiority *p* < 0.05
Sue et al., 2018 [153]	Japan	63	Open-label Multi-center	/	20 mg VPZ twice daily + 750 mg AMX twice daily + 200 mg or 400 mg CLA twice daily for 7 days (VPZ triple therapy for CLA-susceptible and CLA-resistant patients) 30 mg LPZ or 10 mg RPZ or 20 mg EPZ twice daily + 750 mg AMX twice daily + 200 mg or 400 mg CLA twice daily for 7 days (PPI triple therapy for CLA-susceptible patients only)	*In CLA-susceptible patients:* VPZ triple: 87.3% PPI triple: 76.5% *p* = 0.21 *In CLA-resistant patients:* VPZ triple: 82.9% (no comparison with PPI triple)	*In CLA-susceptible patients:* VPZ triple: 88.9% PPI triple: 86.7% *p* = 0.77 *In CLA-resistant patients:* VPZ triple: 82.9% (no comparison with PPI triple)
* In patients with possible/confirmed prior eradication treatment *							
Zuberi et al., 2022 [149]	Pakistan	192	Single-center	Prior treatment status unknown	20 mg VPZ twice daily + 1 g AMX twice daily for 14 days 20 mg OPZ twice daily + 1 g AMX twice daily + 500 mg CLA twice daily for 14 days	NA	VPZ dual: 93.5% PPI triple: 83.9% *p* = 0.042
Sue et al., 2019 [154]	Japan	147	Open-label Single-center	Patients failed first-line (PPI or VPZ + AMX + CLA) and second-line (PPI or VPZ + AMX + MNZ) regimens	20 mg VPZ twice daily + 750 mg AMX twice daily + 100 mg sitafloxacin twice daily for 7 days 30 mg LPZ or 10 mg RPZ or 20 mg EPZ twice daily + 750 mg AMX twice daily + 100 mg sitafloxacin twice daily for 7 days	VPZ group: 75.8% PPI group: 53.3% *p* = 0.071	VPZ group: 83.3% PPI group: 57.1% *p* = 0.043
Hojo et al., 2020 [155]	Japan	46	Open-label Multi-center	Patients failed first-line eradication therapy consisting of VPZ or PPI + AMX + CLA	20 mg VPZ twice daily +750 mg AMX twice daily + 250 mg MNZ twice daily for 7 days 10 mg RPZ twice daily + 750 mg AMX twice daily + 250 mg MNZ twice daily for 7 days	VPZ group: 73.9% PPI group: 82.6% *p* = 0.72	VPZ group: 89.5% PPI group: 86.4% *p* = 1.00
**B.** **Vonoprazan dual or triple therapy vs. PPI-bismuth quadruple therapy**							
Qian et al., 2023 [148]	China	375	Open-label Single-center Noninferiority	/	20 mg VPZ twice daily + 750 mg AMX 4 times daily for 10 days (VHA-dual) 20 mg VPZ twice daily + 750 mg AMX 4 times daily or 1 g AMX twice daily for 10 days (VA-dual) 20 mg EPZ twice daily + 200 mg colloidal bismuth twice daily + 1 g AMX twice daily + 500 mg CLA twice daily for 10 days (BQT)	VHA-dual: 92.7% VA-dual: 84.4% BQT: 89.4% VHA-dual vs. BQT: noninferiority *p* < 0.001 VA-dual vs. BQT: noninferiority *p* > 0.05	VHA-dual: 93.4% VA-dual: 85.4% BQT: 90.9% VHA-dual vs. BQT: noninferiority *p* < 0.001 VA-dual vs. BQT: noninferiority *p* > 0.05
Zhang et al., [156]	China	640	Open-labelr Single-center	/	20 mg VPZ twice daily + 100 mg doxycycline twice daily + 100 mg furazolidone twice daily for 14 days (VDF triple) 20 mg VPZ twice daily + 100 mg doxycycline twice daily + 1 g AMX twice daily for 14 days (VDA triple) 20 mg EPZ twice daily + 220 mg bismuth twice daily + 100 mg doxycycline twice daily + 100 mg furazolidone twice daily for 14 days (EBDF quadruple) 20 mg EPZ twice daily + 220 mg bismuth twice daily + 100 mg doxycycline twice daily + 1 g AMX twice daily for 14 days (EBDA quadruple)	VDF-triple: 88.1% VDA-triple: 87.5% EBDF-quadruple: 80.0% EBDA-quadruple: 75.0% VDF-triple vs. VDA-triple: *p* = 0.864 VDF-triple vs. EBDF-quadruple: *p* = 0.047 VDA-triple vs. EBDA-quadruple: *p* = 0.004 EBDF-quadruple vs. EBDA-quadruple: *p* = 0.284	VDF-triple: 97.9% VDA-triple: 96.6% EBDF-quadruple: 91.4% EBDA-quadruple: 90.2% VDF-triple vs. VDA-triple: *p* = 0.723 VDF-triple vs. EBDF-quadruple: *p* = 0.017 VDA-triple vs. EBDA-quadruple: *p* = 0.049 EBDF-quadruple vs. EBDA-quadruple: *p* = 0.730
**C.** **Vonoprazan-bismuth quadruple therapy vs. PPI-bismuth quadruple therapy**							
Huh et al., 2021 [150]	Korea	30	Double-blind Single-center	Patients were not necessarily treatment-naive	20 mg VPZ twice daily + 1 g AMX twice daily + 500 mg CLA twice daily + 220 mg bismuth twice daily for 14 days; 30 mg LPZ twice daily + 1 g AMX twice daily + 500 mg CLA twice daily + 220 mg bismuth twice daily for 14 days	NA	VPZ quadruple: 100% PPI quadruple: 100%
Lu et al., 2023 [144]	China	234	Open-label Single-center Noninferiority	/	20 mg VPZ twice daily + 1 g AMX twice daily + 100 mg furazolidone twice daily + 200 mg colloidal bismuth twice daily for 10 days (V10 group) or 14 days (V14 group) 20 mg EPZ twice daily + 1 g AMX twice daily + 100 mg furazolidone twice daily + 200 mg colloidal bismuth twice daily for 14 days (E14 group)	V10 group: 96.2% V14 group: 94.9% E14 group: 93.6% V10 vs. E14 group: noninferiority *p* < 0.05 V14 vs. E14 group: noninferiority *p* < 0.05	V10 group: 98.6% V14 group: 97.4% E14 group: 94.8% V10 vs. E14 group: noninferiority *p* < 0.05 V14 vs. E14 group: noninferiority *p* < 0.05
Hou et al., 2022 [147]	China including Taiwan, South Korea	533	Double-blind Multi-center Noninferiority (* primary outcome was duodenal ulcer healing)	All patients had duodenal ulcer (85.4% were *H. pylori*-positive)	In *H. pylori*-positive patients: 20 mg VPZ twice daily + bismuth 600 mg twice daily + 1 g AMX twice daily + 500 mg CLA twice daily for 14 days 30 mg LPZ twice daily + bismuth 600 mg twice daily + 1 g AMX twice daily + 500 mg CLA twice daily for 14 days	VPZ quadruple: 91.5% LPZ quadruple: 86.8% noninferiority *p* < 0.05	NA

* It is to highlight the that unlike other studies, the primary outcome is dudenal ulcer healing instead of HP eradication in this study. Abbreviations: ITT, intention-to-treat analysis; PP: per protocol analysis; CI: confidence interval; PPI, proton pump inhibitor; VPZ, vonoprazan; AMX, amoxicillin; CLA, clarithromycin; LPZ, lansoprazole; OPZ, omeprazole; EPZ, esomeprazole; RPZ, rabeprazole; MNZ, metronidazole; NA, not available.

## Data Availability

No new data were created or analyzed in this study. Data sharing is not applicable to this article.

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
