# Peer review of "Antibiotic Resistance, Susceptibility Testing and Stewardship in *Helicobacter pylori* Infection"

_ijms, 2023, doi:10.3390/ijms241411708_

Round 1
Reviewer 1 Report
1. In Table 1: It will be better to include reference for each resistance mechanism mentioned
2. For better clarity, it will be good to add subheadings to Antibiotic stewardship section
3. Topic selection and general organization of the paper is excellent
4. If possible, reduce / streamline the information in the tables
Only minor edits needed.
Line 181- use 'except' in place of 'expect'
Reviewer 2 Report
1) The abstract section should be updated with an emphasis on the problem statement and novelty of the work.
2) The introduction section requires improvement because it is so brief. It should compile critical findings from the preceding studies and highlight the research gaps that motivate authors to carry on with this study.
3) I think at least 2-3 figures should be added in the paper because figures are the beauty of a review paper to describe the whole materials/procedure/process, some interesting self based figures (with own idea) should be added to desceibe the whole process.
4) How can we raise public awareness for antibiotic resistance?
5) How can healthcare providers help prevent the spread of antibiotic resistance? What are ways that patients can prevent antibiotic resistance in healthcare settings?
6) Discus the limitations and the strengths of this study.
7) The conclusion section is not enough and needs to be upgraded.
Nil
Reviewer 3 Report
In this study, the authors gave an overview of the current H. pylori antibiotic resistance patterns and mechanisms as well as different antibiotic susceptibility testing methods that are currently in use.
You have no keywords. Please add them.
You did not specify what type of study it is. Please add this information.
Please explain all abbreviations that appear in the article (eg AST).
The 4 tables are clear.
The article presents 158 references, being up to date. In my opinion, the 7 self-citations you used are too many. I recommend you to reduce their number.
Round 2
Reviewer 2 Report
I have critically and carefully evaluated this paper and the authors gave careful consideration to every comment. The paper is now suitable for publication.
NA